Optimizing multimodal feature selection using binary reinforced cuckoo search algorithm for improved classification performance

Thirugnanasambandam Kalaipriyan 1
http://orcid.org/0000-0002-7297-9161 Murugan Jayalakshmi 2
http://orcid.org/0000-0002-0734-3423 Ramalingam Rajakumar 3
Rashid Mamoon 4 mamoon873@gmail.com
Raghav R. S. 5
Kim Tai-hoon 6 taihoonn@chonnam.ac.kr
Sampedro Gabriel Avelino 7 8
Abisado Mideth 9
1 Centre for Smart Grid Technologies, School of Computer Science and Engineering, Vellore Institute of Technology , Chennai , India
2 Department of Computer Science and Engineering, Kalasalingam Academy of Research and Education , Krishnankoil , India
3 Centre for Automation, School of Computer Science and Engineering, Vellore Institute of Technology , Chennai , India
4 Department of Computer Engineering, Faculty of Science and Technology, Vishwakarma University , Pune , India
5 School of Computing, SASTRA Deemed University , Villupuram , India
6 School of Electrical and Computer Engineering, Chonnam National University , Daehak-7 , Republic of Korea
7 Faculty of Information and Communication Studies, University of the Philippines Open University , Los Baños , Philippines
8 Center for Computational Imaging and Visual Innovations, De La Salle University , Malate , Philippines
9 College of Computing and Information Technologies, National University , Manila , Philippines
Alatas Bilal
Electronic publication date: 2024 Jan 29
Publication date: 2024
Volume: 10
Electronic Location ID: e1816
Received 2023 Aug 16; Accepted 2023 Dec 19
Copyright: © 2024 Thirugnanasambandam et al.
Copyright year: 2024
Copyright holder: Thirugnanasambandam et al.
License: This is an open access article distributed under the terms of the Creative Commons Attribution License, which permits unrestricted use, distribution, reproduction and adaptation in any medium and for any purpose provided that it is properly attributed. For attribution, the original author(s), title, publication source (PeerJ Computer Science) and either DOI or URL of the article must be cited.
License URL: https://creativecommons.org/licenses/by/4.0/

Keywords: Reinforced cuckoo search, Multimodal, Binary solution space, Feature selection, Machine learning, Artificial intelligence, Emerging technologies, Data science

Funding: The authors received no funding for this work.

==============================
Background

Feature selection is a vital process in data mining and machine learning approaches by determining which characteristics, out of the available features, are most appropriate for categorization or knowledge representation. However, the challenging task is finding a chosen subset of elements from a given set of features to represent or extract knowledge from raw data. The number of features selected should be appropriately limited and substantial to prevent results from deviating from accuracy. When it comes to the computational time cost, feature selection is crucial. A feature selection model is put out in this study to address the feature selection issue concerning multimodal.

Methods

In this work, a novel optimization algorithm inspired by cuckoo birds’ behavior is the Binary Reinforced Cuckoo Search Algorithm (BRCSA). In addition, we applied the proposed BRCSA-based classification approach for multimodal feature selection. The proposed method aims to select the most relevant features from multiple modalities to improve the model’s classification performance. The BRCSA algorithm is used to optimize the feature selection process, and a binary encoding scheme is employed to represent the selected features.

Results

The experiments are conducted on several benchmark datasets, and the results are compared with other state-of-the-art feature selection methods to evaluate the effectiveness of the proposed method. The experimental results demonstrate that the proposed BRCSA-based approach outperforms other methods in terms of classification accuracy, indicating its potential applicability in real-world applications. In specific on accuracy of classification (average), the proposed algorithm outperforms the existing methods such as DGUFS with 32%, MBOICO with 24%, MBOLF with 29%, WOASAT 22%, BGSA with 28%, HGSA 39%, FS-BGSK 37%, FS-pBGSK 42%, and BSSA 40%.

Introduction

In data mining, feature extraction and feature selection are two significant phases that are vital in dimensionality reduction. Feature selection technique aids in various machine learning approaches by choosing a subset of features from the collection of available features to improve classification accuracy. Since entire computing features take a lot of time, feature selection is now a common practice in pattern representation. In addition to these drawbacks, it also considerably slows down the learning process (Unler, Murat & Chinnam, 2011). The over-fitting of the training data, which results in an incorrect perception of a learning algorithm with inappropriate and repetitive characteristics, can occasionally drop classification accuracy. Implementing feature selection would be appropriate while creating a pattern classifier to speed up computation and increase classification accuracy. The primary insights in each pattern, such as noise reduction, are examined when there are less convenient features for pattern categorization (Wang et al., 2007).

According to Yang & Olafsson (2006), the feature selection problem is an entirely problematic concept to solve since it is based on the combinatorial problem. Due to this, the feature selection problem is now viewed as a discrete optimization problem that may be effectively and efficiently solved using a wide range of machine learning and optimization techniques such as support vector machines (Maldonado & Weber, 2009), genetic algorithm (Yang & Honavar, 1998), evolutionary (Bermejo, Gámez & Puerta, 2011) and unsupervised learning approaches (Kim, Street & Menczer, 2000). In the optimization field, evolutionary algorithms (EA) recently showed a substantial result that a conventional optimization strategy fails to achieve. Due to its efficient investigation of a given solution space through population-based search methods, evolutionary approaches have a high success rate.

Multi-objective optimization problems can be solved when a problem’s solution space has multiple best solutions for a single objective. The algorithm’s flow is complicated when multiple optimal solution from a search space is treated in a single run. The literature discusses niche approaches specially created for expanding evolution-based algorithms for convergent convergence of the individual under various peaks. These approaches enable parallel convergence towards numerous peak solutions in the given multimodal search space. Niching is a concept that has its roots in nature. By adopting the niche strategy, a population will be divided into several subpopulations, each operating at a different peak to produce numerous solutions while maintaining the system’s optimum performance.

The objective of feature selection is to select the minor features from a dataset or pattern that adequately captures the learning process. Furthermore, not all characteristics of a design need to be distinctive. Therefore, from a multimodal optimization (MO) standpoint, feature selection aims to acquire the maximum subset with the least number of characteristics in each without sacrificing accuracy. MO’s increased exploration and exploitation power allow it to converge towards different solutions. In feature selection from a vast solution space, several subsets of a few features should be obtained. As a result, the feature selection problem can be expressed as a non-deterministic polynomial-complete problem, where the number of input features is discrete. Still, the computational effort grows exponentially with the number of features. Standard EA provides an ideal answer from a broad search space for a given issue. The use of MO approaches increases when the best solution found cannot be used in real-world situations or when the best solution is expensive to implement. When a customer is presented with a more significant number of optimum solutions rather than just one, they can choose a solution that satisfies their restrictions without sacrificing optimality.

In the last 10 years, many optimization issues have been solved using various optimization algorithms. Though, there are a lot of current significant evolutionary methods available in literature. In this work, we used Cuckoo Search algorithm (CSA) to address the multimodal feature selection issue. The cuckoo search method was chosen because of its independent brooding parasitism behavior, detailed in Thirugnanasambandam et al. (2019). We developed the proposed Binary Reinforced Cuckoo Search Algorithm (BRCSA) to address multimodal optimization in feature selection.

The primary contributions of this work are listed below. A practical Binary Novel External Archive is proposed to hold multiple solutions from the same optimal pool of solutions.

Additionally, two distinct Boolean operators were added to the continuous Reinforced Cuckoo Search Algorithm (RCSA) to handle the binary solutions.

Unlike standard CSA, the reinforced CSA algorithm is capable to eradicate the local optimal struck and explores the search space within the boundary region.

To validate the efficacy of the proposed BRCSA approach, an extensive evaluation with the recent approaches is determined to prove the performance of the proposed algorithm.

The rest of the article is structured as follows: “Related Work” analyses relevant studies in the feature selection field to pinpoint research gaps. The Binary Novel External Archive, which recognizes and stores the solutions under various peaks, is covered in “External Binary Novel Archive”. In “Proposed Binary Reinforced Cuckoo Search Algorithm”, the suggested BRCSA algorithm is discussed. The thorough experimental study of the recommended strategy was presented in “Experimentation and Result Analysis”. The article’s future enhancement potential is covered in “Results Analysis”.

Related work

In data processing, there are numerous mathematical optimization models, metaheuristic algorithm-based optimization models, and deterministic methods to handle feature selection. The feature selection problem’s current approaches to solving it and its research gaps were covered in this section. The plans in evolutionary algorithms for maintaining multimodal solutions have also been discussed.

Evolutionary strategy (ES) (Schwefel, 1977) is influenced by the idea that evolution changes over time. Each solution in the population of ES contains a set of tactical factors and decision variables that need to be modified or optimized. The strategic parameters and choice variables are tuned at the time of evolution, and the strategic parameters influence the change in decision variables. Currently, the mutation point in continuous search space is handled by adding a variable to the existing individual.

Shir & Bäck (2005a) published a dynamic niching approach based on ES in 2005; the first algorithm considered to incorporate the niching method in ES. The first phase in this process is randomly mutating everyone using a self-adaptive technique, after which each individual’s fitness value is calculated. Each peak individual is monitored in ES based on its fitness value using dynamic peak identification (Miller & Shaw, 1996). Calculating a dynamic niche radius value using the individual allocated to each peak is necessary. A mating restriction is implemented during this stage, requiring each dominant individual to mate only with other group members. With a standard mating norm, each niche can generate a group of children. This approach severely restricts the strong dominance of the entire population by a single niche individual.

In 2005, Shir & Bäck (2005b) presented the Covariance Matrix Adaptation based ES (CMA-ES), a hybrid ES and dynamic niching technology. According to its run-time search procedure, this method adapts the prior algorithm. Actual matrix variance adaptation and cumulative step adaptation are the two stages used for adaptation. The second technique is defined as the control over the evolution path, whereas the first adaptation mechanism uses the method to regulate the overall length of the track. This approach uses the data from the phase of consecutive iteration mutation.

Shir et al. (2005) demonstrated the effectiveness of ES-based niching approaches in 2006 when applied to the quantum control problem with second laser pulse shaping. The methods for multimodal optimization have been thoroughly studied in this article. Comparing the results with the existing single population-based mechanism, which requires multiple runs to obtain optimal solutions from different peaks, reveals a significant performance improvement.

Fixing the niche radius in the past is difficult when the search space has not been well investigated. In multimodal optimization, selecting such parameters before running an algorithm is complicated. Shir & Bäck (2006) presented a self-adaptive system between 2006 and Shir, Emmerich & Bäck (2010) where each individual’s niche radius will be decided. Two alternative techniques are proposed to identify respective radii, the first of which uses Mahalabonis distant metrics and the second of which uses cumulative step size adaptation. A stand-alone mechanism has been developed using intricate geometrical structures to determine the niche radii for those already acting as niche seeds. The fact that additional parameter specifications come along with this adaptation mechanism and increase the complexity of an algorithm by allowing for the tuning of more parameters is one of the drawbacks of this approach to tackling multimodal optimization problems.

An ensemble constrained Laplacian score based feature selection model (CLS) was proposed by Benabdeslem, Elghazel & Hindawi (2016). A semi-supervised feature selection approach is developed to remove the scan supervision information from the Laplacian constraint score. The suggested process eliminates the restriction by enforcing data resampling and a random subspace method. To solve the feature selection problem, Mafarja & Mirjalili (2017) introduced a hybrid bio-inspired optimization model dubbed the Whale Optimization algorithm with a heuristic method called Simulated Annealing (WOASAT). The simulated annealing model improves the efficiency of the Whale optimization algorithm to identify the best solutions. When the results are compared to current methods, accuracy scores are prevalent.

Guo & Zhu (2018) introduced a dependence-guided unsupervised feature selection model (DGUFS) in 2018 to replace the two-step sparse learning process of feature selection and clustering. The suggested model uses an alternate direction method for multipliers to solve the minimum constraint problem to address this strategy (Faris et al., 2018) suggested a binary-based salp swarm optimization algorithm with an improved operator for crossover operation to increase the exploration probability while looking for nearly optimum solutions. The outcomes demonstrate that this model performs better and produces useful findings across many datasets.

In 2019, Mafarja et al. (2019) developed the binary grasshopper optimization technique, another bio-inspired-based optimization model for selecting the proper characteristics to address the UCI dataset issues. The suggested model enhances neighbourhood selection as the problem involves more local optima. The continuous optimization approach is converted to address binary issues, and V-shaped and sigmoid transfer functions are applied. A mutant operator was additionally added to improve the exploitation process. Analysis of the results reveals a notable increase in accuracy compared to other models.

To tackle feature selection challenges, Alweshah et al. (2021) suggested a hybrid model of the Mine blast technique with simulated annealing in 2020. The mine blast algorithm is used in the proposed model to enhance the exploration of feature selection search space. Additionally, to improve the exploitation during searching, simulated annealing is applied. HGSA outperforms other cutting-edge methods in testing and results comparisons with current models. Another Wrapper-based Feature selection approaches employs k-nearest neighbour, and the same author proposed the monarch butterfly optimization algorithm in the same year (Alweshah, 2021). An improved crossover operator is used in the first model MBOICO to improve the k-nearest neighbour classification technique. Levy flying distribution is used in the second model MBOLF to enhance performance in terms of exploration.

Binary gaining-sharing knowledge algorithm-based knowledge-based models were proposed by Agrawal, Ganesh & Mohamed (2020). The suggested sharing approach has two distinct levels: beginning-level sharing and expert-level sharing. The proposed models come in two different iterations, FS-BGSK and FS-pBGSK. The first model uses the information mentioned above sharing, and the FS-pBGSK population reduction strategy is enforced to improve exploration performance.

In 2021, Hu et al. (2021) proposed multimodal PSO for solving feature selection problems. The algorithm uses the Hamming distance to determine how far apart any two particles are. The two suggested variations of MNPSO, MNPSO-C (using crowding clustering) and MNPSO-S (using speciation clustering), utilize two niching update techniques for multimodal feature selection. The best particle in the niche, rather than the best particle overall, is used to inform velocity updates to improve communication amongst particles in the same place. The feature sets that provide the best classification accuracy are archived in a separate repository. In 2022, Wang, Li & Chen (2022) proposed a genetic algorithm-based multimodal feature selection algorithm from a multi-objective prospect which uses a unique crowding distance computation method. Particular calculations for crowding distance may take the variety of the decision space and the diversity of the object space into account. Combining the simulated binary crossover (SBX) approach with the ability of Pareto solutions to create child solutions yields a novel crossover mechanism. The equilibrium between convergence and variety in the decision and object space may be ensured simultaneously, and both PS distribution and PF precision can be improved.

In 2023, Agrawal et al. (2023) proposed a multi-objective-oriented feature subset selection that uses the niching technique for identifying the non-dominated solutions. The authors suggested the probability initialization approach for identifying features with equal distribution in the search space. In addition, the model holds a niching strategy for navigating the search space and exploiting neighboring solutions. Also, it is presented with a convergence archive for locating and storing the optimum feature subsets. On the other hand, soft computing, machine learning, deep learning, and optimization algorithms have a wide range of applications addressed in various domains, such as the Internet of Things (IoT), Big Data, etc. Among the models are those for linguistic steganography compression (Xiang et al., 2018), fetal plane recognition (Pu et al., 2021), traffic flow prediction (Chen et al., 2020), various Big Data service architectures (Wang et al., 2020), and blockchain systems (Zhang et al., 2020).

In light of the aforementioned advantages of memetic algorithms in the context of feature selection, it is pertinent to consider the necessity of further novel memetic approaches. In the domain of optimization, there exists a theory known as the No-Free-Lunch (NFL) theorem, which provides logical proof that the existence of a universal algorithm capable of addressing all optimization problems is unattainable. Within the context of this study, it can be asserted that none of the heuristic wrapper feature selection methods possess the capability to effectively address all feature selection challenges. In alternative terms, there exists a continuous scope for enhancing the existing methodologies in order to more effectively address the present challenges pertaining to feature selection. The motivation behind our efforts is to present an additional memetic method for feature selection in the subsequent part.

The recent advancements on evolutionary algorithms and its application domains includes ship trajectory prediction using evolutionary strategy by Zheng et al. (2023) and using long short-term memory by Qian et al. (2022). On solving flow shop scheduling problem, Lu et al. (2023) used iterated greedy search method and Zheng et al. (2022) used neural network-based track prediction method. The evolutionary algorithms and deep learning methods plays a vital role in image processing domain where for identifying image captioning Wang et al. (2022) used interaction learning, for image and sentence matching Li et al. (2023) used bi-directional aggregation. The other different domain of imposing AI strategies is gripper object interaction (She et al., 2022), temperature impedance sensor design (Huang et al., 2023), image fusion using multiscale feature extraction (Lu et al., 2023) sparse decomposition (Qin et al., 2022) and so on.

External binary novel archive

The solutions of various peaks will be recognized and saved in the External archive, a solution-preserving storage area in evolutionary computation. The differences between solutions of various peaks are discovered to utilize a sample solution’s feasibility check after an initial conditional examination. Creating a sample solution that falls between the two ideal solutions and checking the constraint mapping makes it possible to detect differences in continuous solution space. If the solution does not follow the constraint, the two individuals are from different peaks. The same strategy, however, cannot be applied to solutions with binary representation because an individual with binary representation cannot use such a generating method. Therefore, a Binary Novel External Archive with low computational cost has been adopted in this article.

The Binary Novel External Archive uses the steps below to locate and save solutions from various peaks.

Step 1: The best solution from the first iteration will be inserted into the external archive without any conditional checks or references.

Step 2: The fitness value of the optimal solution will be compared to each member of the existing population. The individual will exceed the individual to the non-similarity check when the Hamming Distance between the individuals is more prominent.

Step 3: This dissimilarity check will involve input from all individual variables. It will be counted how many different variables there are overall between each individual.

Step 4: A conditional dissimilarity check will be performed, determining whether the solution can be placed into the external archive if the proportion of different variables to the total variables is greater than 50%.

Step 5: Each member of the current population and each population in the external archive will engage in this process.

In Algorithm 1, the pseudocode for the proposed Binary External Archive has been briefly illustrated. The current iteration number, hamming distance, external archive, and individual addition to the external library are all represented in this algorithm. The annotations used in Algorithms 1 and 2 are displayed in Table 1.

Algorithm 1 Binary Novel External Archive.

Input: Candidate solutions (x), Objective function (f)	
Output: Binary Novel External Archive	
Initialize Flag = 0; Υ=0	
if ( iter=1) do	
   Arh←xbest	
end if	
∀i∈pop do	
   ∀k∈Arh do	
    if ( HD(f(xi),f(xbest))>0) then	
       ∀j∈D do	
        if (xi,j≠xk,j) then	
           Υ←Υ+1	
        end if	
      end for	
    end if	
    if (Υ≥D+12) then	
      Flag=1	
       exit()	
    end if	
  end for	
  if (Flag≠1), then	
     Arh← Arh ⊕xi	
    Flag=0	
  end if	
end for	

Algorithm 2 Binary Reinforced Cuckoo Search Algorithm.

Input: Total Population of the nest ( N), dimensions (d), objective function f()	
Output: Fitness_xbest and xbest	
Each x consists of dimensions (1… d)	
PopNest, Termination Criteria, α=0.1,Pa	
1: Initialize individuals in host nest x	
2: for i←1toN do	
3:   xi←randi([0,1],d,1)	
4:   Calculate Fitness_xi←f(xi)	
5: end for	
6: while (Termination Condition not met) do	
7:     for i←1toN	
9:         Levy=randi([0,1],1,1)	
10:      xt!←randi([0,1],d,1)	
11:       xt2←randi([0,1],d,1)	
12:       yi = xt!⋈xt2	
13:    end for	
14:   Calculate Fitness_yi←f(yi) where i∈1…N	
15:   for i←1toN	
16:      if ( Fitness_yi≤Fitness_xi)	
17:          xi←yi	
18:          Fitness_xi←Fitness_yi	
19:     end if	
20:  end for	
21:  Compute mean value σt=∑i=1N⁡f(xi)N	
22:  Compute Fitness Difference ϑt=C×(σt−f(xtbest))	
23:  Compute Threshold Value Θt=f(xtbest)+ϑt	
24:  Find the index of abandon solutions A∘t,k=index(f(xi))>Θt|∀andk∈{1	
25:  Find the index of qualified solutions Qt,L=A∘t,k	
26:  Calculate α=|f(xtbest)σt×ω|φ	
27:  for i=1tosize(A∘t,k), then	
28:        ΔX=∑j=1m⁡(xj−xi)2m where m∈Qt,K	
29:       if α>0.5,then	
30:          xi=Sum(xnew⊕ΔX)	
31:       else	
32:          xi=Carry(xnew⊕ΔX)	
33:       end if	
34:  end for	
35:   Fitness_xbest=min(f(xi))	
36:   xbest=index(Fitness_xbest)	
37: end while	

Table 1 Annotations used in the proposed algorithms.

Parameter term	Notations	
Individual solution	x	
#Individuals in a population	N	
Fitness function	f(x)	
Dimension	D	
Archived solutions	Arh	
Hamming distance	HD	
Best individual solution	xbest	
Multiplexer coefficient (Pseudo-random)	S0	
Multiplexer operation	⋈	
Mean of entire population fitness values	σ	
Abandoned solutions	Å	
Qualified solutions	Q	
Rate of deviation between qualified solutions and abandoned solution	ΔX	

Proposed binary reinforced cuckoo search algorithm

The Reinforced Cuckoo Search Algorithm (RCSA) was created to address continuous non-linear problems where the design variables are chosen in a way that allows the RCSA to search in the given search space efficiently. Since the solution representation for the feature selection problem is binary, it is incorrect to address the same RCSA when solving it. The variables must be created so that they can only hold the values 1 or 0, indicating whether the feature has been chosen to participate in the objective function. Therefore, the proposed BRCSA must be modified to include binary solution representation.

On the other hand, several standard modules, such as the Transfer function and binarization (Kennedy & Eberhart, 1997), Great Value Priority and Mapping (Crawford et al., 2017), and the Angle Modulation Rule (Proakis et al., 1994), exist to produce the metaheuristic algorithms for solving the binary-represented search space. These are the three well-known mathematical methods for the binary transformation of a set of continuous solutions. We deduced a model based on Boolean operators to handle the binary solution representation in our model without jeopardizing the integrity of RCSA’s work.

Fitness function for feature selection

The fitness function is critical in evaluating candidate solutions and directing the search process towards determining the optimal feature subsets. Sebban & Nock (2002) classified the feature selection approach into three major classes based on what needed to be optimized. Out of three major classes, we selected the third classes which deals the classification accuracy as the main discriminability computation to attain the maximum precision value. Hence, the mathematical formulation of fitness function for evaluating the selected feature subset is expressed in Eq. (1):

(1) Fitness(xi)=Ψ(Acc(FS))+(1−Ψ)(FN−FSFN)

where xi denotes the candidate solution with selected features. Acc(FS) specifies the classification accuracy of selected FS features. FN and FS symbolizes the total number of features and selected number of features, respectively. Ψ denotes the constant coefficient aids to tradeoff the effect of Acc and FN in computing fitness value. Based on the experimentation, we fixed the Ψ value as 0.6. Fitness function aids the algorithm towards the best accuracy by utilizing a smaller number of features in the search boundary.

Algorithm for binary reinforced cuckoo search (BRCSA)

This section deliberates the proposed BRCSA pseudocode and the incorporation of the Multiplexer and Binary Adder models into the BRCA algorithm. The main motive of introducing reinforced cuckoo search is to eradicate the local optimal struck and premature convergence. The proposed model is capable to overcome the issues mentioned above as well as improves the trade-off between exploration and exploitation process. The detailed procedure of BRCA is provided in Algorithm 2.

Experimentation and result analysis

The performance of the suggested method has been examined and implemented on 11 different datasets with various features retrieved from the UCI Machine Repository (UC Irvine Machine Learning Repository). These datasets are found to be accessed and utilized for assessment and comparison purposes. The suggested strategy has now been contrasted with well-known heuristic methods listed in the following readings.

Datasets description

The datasets as mentioned above are used to compare the performance of the proposed method with other existing methods. Wisconsin Breast Cancer (WBCO), Zoo, Glass, Protein, Wdbc, Lung Cancer, Wine, Vehicle, Segment, Soybean, and Sonar are among the 11 datasets contained in the dataset. The datasets are chosen in accordance with the references provided by cutting-edge algorithms to support their proposed feature selection mechanism. It is made up of large samples with various attributes. Furthermore, the feature selection will yield results with a multimodal aspect.

Evaluation mechanism

The accuracy of the classification and the total number of features used to calculate the tabular accuracy level are compared between the results of the proposed method and other current methods. The cutting-edge algorithms were utilized to assess the effectiveness of the suggested strategy DGUFS (Guo & Zhu, 2018), MBOICO (Alweshah, 2021), MBOLF (Alweshah, 2021), WOASAT (Guo & Zhu, 2018), BGSA (Mafarja et al., 2019), HGSA (Alweshah et al., 2021), FS-BGSK (Agrawal, Ganesh & Mohamed, 2020), FS-pBGSK (Agrawal, Ganesh & Mohamed, 2020), and BSSA (Faris et al., 2018) are the algorithms that have been selected for comparison. The algorithm dependent parameters are fixed for the proposed algorithm based on the empirical results. The algorithm dependent parameters of the respective algorithms are chosen from the respective research articles. The Population size, maximum number of iterations, number of runs, number of decision variables to be optimized for all these algorithms are fixed as mentioned in the proposed algorithm parameter values Table 2. These algorithms were chosen for comparison because, in recent years, their efficacy in feature selection has been comparable.

Table 2 Parameter settings of the proposed algorithm.

Sl. No.	Parameters	Values	
1	Population size	100	
2	Total number of iterations	1,000	
3	Dimensions	Based on dataset	
4	Runs	10	
5	α	0.1	
6	Pa	0.25	

Each dataset was divided into three equal halves for testing and training. In other words, 40% of each dataset’s instances are used for testing, while 60% of each dataset’s samples are used to train the nearest neighbor (1-NN) classifier. The proposed model is evaluated on the computational device with the configuration Intel Core i7 4.2 GHz of processor speed, 8 GB primary memory and 2 TB secondary memory. The proposed and existing algorithms are implemented in MATLAB version 9 (The MathWorks, Inc., 2022).

Results analysis

The results of various multimodal optimization techniques on the 11 datasets, each with a different number of attributes, are tabulated in Table 3. The individual algorithms’ classification accuracy (CA) was trained and evaluated using a 1-NN classifier. The mean and standard deviation of the selected features and the classification accuracy is shown in the table. The outcome demonstrates the effectiveness of the natural memetic algorithm, which outperforms existing MO algorithms in obtaining close to optimal solutions for multimodal optimization. The similarity between the realistic effects of Memetic and MO approaches led to the natural selection of the MO-based treatment. Although DFS occasionally achieves ideal results, there is no discernible performance compared to WOASAT. The exploitation tactic it employs to look for the neighbourhood is the root of this degeneration. It will lead us to believe that the exploitation phase is crucial to feature selection with multimodal characteristics.

Table 3 Results on the average number of features of BRCSA vs existing algorithms.

Dataset	WBCO	Glass	Wine	Zoo	Vehicle	Protein	Segment	Wdbc	Soybean	Lung cancer	Sonar	
DFS	2.05	3.4	2.65	4.6	5.6	7.75	3.3	4.95	4.75	19.4	18.75	
MBOICO	12	3.4	2.7	3.9	4.8	7.25	3	2.55	2	9.35	12.2	
MBOLF	2.75	3.35	2.55	4.5	4.8	7.95	3	2.9	1.9	7.9	11.85	
FS-BGSK	1.45	3.5	2.6	4.75	5.15	7.95	3	2.6	2	13.1	14.4	
BGSA	1.67	2.88	2.47	3.8	5.01	6.98	3	2.58	2.55	18.1	14.1	
FS-pBGSK	1.55	3.4	2.75	4.7	5.05	8.3	3	2.75	1.9	12	12.6	
HGSA	1.6	3.8	4.1	5.3	5.4	8.7	3.85	3.15	2.25	13.7	16.4	
DGUFS	1.5	3.65	2.8	5.9	5.5	8.2	3	2.5	2	7.9	12.15	
WOASAT	3	4.6	2	4.8	5.6	7.4	3.4	4.8	7.05	18.8	18.8	
BSSA	2.8	4.6	2.4	6.6	5	7.4	3	2.4	4.95	15.6	12.8	
BRCSA	1.4	3.3	2.55	4.5	4.6	7.2	3	2.5	1.9	14.1	12	
Without FS	9	9	13	16	19	20	20	30	35	56	60	

The proposed method will generally only be evaluated on single modal version-based datasets when it outperforms or is nearly equivalent to each other in one of the several instances. The standard difference between the efficacy of MO approaches and different single modal algorithms will be effectively differentiated when classification on this dataset is performed. On this dataset, MO algorithms like HGSA find the correct number of characteristics to categorize the data set among the various options that are already available. It has been accomplished using the algorithm’s effective exploration technique.

The average number of features the algorithms yielded throughout all 20 runs is displayed in Table 2. Figure 1 compares each dataset’s average number of elements obtained to achieve the highest classification accuracy overall.

Figure 1 Comparison of average features for different instances.

Analysis of the results for the median number of features

In this part, the effectiveness of the suggested algorithm has been examined using several datasets that contain varying numbers of characteristics. Table 4 and Fig. 2 illustrates the classification accuracy for various instances obtained by proposed method and other compared techniques. The outcome of proposed model with other comparative models on various datasets are discussed separately. The justification of proposed method outcome is provided below.

Table 4 Results on average classification accuracy of RCSA vs exiting algorithms.

Dataset	WBCO	Glass	Wine	Zoo	Vehicle	Protein	Segment	Wdbc	Soybean	Lung cancer	Sonar	
DFS	0.9337	0.7157	0.9514	0.9687	0.7336	0.788	0.9674	0.9342	1	0.9269	0.9265	
MBOICO	0.934	0.7186	0.9563	0.9775	0.727	0.8173	0.9665	0.9418	1	0.9846	0.9668	
MBOLF	0.8741	0.7104	0.95	0.965	0.7302	0.8348	0.9693	0.9432	1	0.9846	0.9674	
FS-BGSK	0.939	0.786	0.9563	0.9775	0.7303	0.8195	0.966	0.9381	1	0.95	0.951	
BGSA	0.9387	0.7535	0.9537	0.961	0.7337	0.7826	0.9751	0.9342	1	0.9538	0.9324	
FS-pBGSK	0.941	0.7462	0.9605	0.9787	0.7363	0.8304	0.9706	0.9394	1	0.98	0.946	
HGSA	0.9306	0.6973	0.9114	0.9725	0.7225	0.7923	0.9301	0.9399	1	0.95	0.9502	
DGUFS	0.9282	0.6773	0.9394	0.9662	0.7312	0.8141	0.9662	0.9373	1	0.9961	0.9583	
WOASAT	0.9247	0.6359	0.9239	0.965	0.7219	0.8087	0.9507	0.9298	1	0.9692	0.9181	
BSSA	0.9311	0.6767	0.938	0.97	0.697	0.8174	0.969	0.9281	1	0.9385	0.9398	
BRCSA	0.942	0.819	0.96	0.985	0.731	0.843	0.966	0.951	1	0.9962	0.97	
Without FS	0.96	0.671	0.732	0.932	0.69	0.691	0.96	0.916	0.984	0.461	0.816	

Figure 2 Comparison of classification accuracy for several instances.

For the WBCO dataset, the proposed approach outperformed the following algorithms: DFS (32%), MBOICO (88%), MBOLF (49%), FS-BGSK (3%), BGSA (16%), FS-pBGSK (10%), HGSA (13%), DGUFS (7%), and WOASAT (53%).

For the Glass dataset, the suggested approach performs better than other algorithms like DFS, MBOICO, MBOLF, FS-BGSK, FS-pBGSK, HGSA, DGUFS, and WOASAT, which all perform worse by 3%, 6%, 3%, 10%, and 1%, respectively. However, the proposed algorithm’s performance versus BGSA has decreased by −15%. For the Wine dataset, the proposed approach outperformed the following algorithms: DFS (4%), MBOICO (6%), MBOLF (0%), FS-BGSK (2%), FS-pBGSK (7%), HGSA (38%), and DGUFS (9%). However, the suggested approach is outperformed by BGSA and WOASAT, which locate an average number of features with 3% and 28% better performance, respectively.

For the Zoo dataset, the suggested approach performs better than the current algorithms DFS (2%), FS-BGSK (5%), FS-pBGSK (4%), HGSA (15%), DGUFS (24%), and WOASAT (6%). With a similar number of average feature sets, the proposed BRCSA competes with MBOLF. However, the suggested method is outperformed by MBOICO and BGSA, which locate 15% and 18% more features than it does. For the Vehicle dataset, the proposed algorithm outperformed DFS (18%), MBOICO (4%), MBOLF (4%), FS-BGSK (11%), BGSA (8%), FS-pBGSK (9%), HGSA (15%), DGUFS (16%), and WOASAT (18%). For the Protein dataset, the suggested method outperforms the currently used algorithms DFS (7%), MBOICO (1%), MBOLF (9%), FS-BGSK (9%), FS-pBGSK (13%), HGSA (17%), DGUFS (12%), and WOASAT (3%). However, the proposed approach is outperformed by BGSA, which finds 3% more features than the proposed algorithm.

For the Segment dataset, the proposed algorithm outperformed DFS by 9%, HGSA by 22%, and WOASAT by 12% in performance. The suggested BRCSA is similarly competitive with MBOICO, MBOLF, FS-BGSK, BGSA, FS-pBGSK, and DGUFS. For the Wbdc dataset, the proposed approach performs better than other algorithms like DFS (49%), MBOICO (2%), MBOLF (14%), FS-BGSK (4%), BGSA (3%), FS-pBGSK (9%), HGSA (21%), and WOASAT (48%). Additionally, it suggested BRCSA competes with DGUFS on discovering the fewest possible feature sets.

The proposed algorithm outperformed DFS with 60%, MBOICO with 5%, FS-BGSK with 5%, BGSA with 25%, HGSA with 16%, DGUFS with 5%, and WOASAT with 73% for the Soyabean dataset. However, when identifying an average number of basic feature sets to categorize the provided ingredient, the suggested BRCSA competes on an equal footing with MBOLF and FS-pBGSK. For the Wbdc dataset, the suggested approach performs better than the already used techniques, including DFS (27%), BGSA (22%), and WOASAT (25%). However, the performance of the suggested method is significantly outperformed by MBOICO, MBOLF, FS-BGSK, FS-pBGSK, and HGSA.

For the Sonar dataset, the proposed algorithm outperformed DFS (36%), MBOICO (2%), FS-BGSK (17%), BGSA (15%), FS-pBGSK (5%), HGSA (27%), DGUFS (1%), and WOASAT (36%). MBOLF, however, outperforms the suggested BRCSA with 1% better outcomes. The proposed method outperformed DFS with 1%, MBOICO with 1%, MBOLF with 7%, HGSA with 1%, DGUFS with 1%, and WOASAT with 2% for the WBCO dataset. Additionally, it is in direct competition with already-existing algorithms like FS-BGSK, BGSA, and FS-pBGSK.

On the Glass dataset, the proposed approach performs better than other algorithms like DFS (13%), MBOICO (12%), MBOLF (13%), FS-BGSK (4%), BGSA (8%), FS-pBGSK (9%), HGSA (15%), DGUFS (17%), and WOASAT (22%). The suggested algorithm outperformed DFS with 1%, MBOICO with 0%, MBOLF with 1%, BGSA with 1%, HGSA with 5%, DGUFS with 2%, and WOASAT with 4% for the Wine dataset. Additionally, it is in direct competition with already-existing algorithms like FS-BGSK and FS-pBGSK. The proposed approach outperforms known algorithms for the Zoo dataset, including DFS by 2%, MBOICO by 1%, MBOLF by 2%, FS-BGSK by 1%, BGSA by 2%, FS-pBGSK by 1%, HGSA by 1%, DGUFS by 2%, and WOASAT by 2%.

For the Vehicle dataset, the suggested algorithm outperformed MBOICO, HGSA, and WOASAT by 1%, 1%, and 1%, respectively. Additionally, it engages in equal competition with DFS, MBOLF, FS-BGSK, BGSA, and DGUFS. To classify the dataset effectively, FS-pBGSK surpasses BRCSA with a 1% improved outcome in terms of classification accuracy. For the Protein dataset, the proposed algorithm performs better than the existing algorithms DFS (7%), MBOICO (3%), MBOLF (1%), FS-BGSK (3%), BGSA (7%), FS-pBGSK (1%), HGSA (6%), DGUFS (3%), and WOASAT (4%). For the Segment dataset, the suggested approach outperformed HGSA and WOASAT by 4% and 2%, respectively. Additionally, it engages in equal competition with DFS, MBOICO, MBOLF, FS-BGSK, FS-pBGSK, and DGUFS. To efficiently categorize the dataset, BGSA outperforms BRCSA with a 1% enhanced classification accuracy result.

For the Wbdc dataset, the suggested approach performs better than the current algorithms DFS (2%), MBOICO (1%), MBOLF (1%), FS-BGSK (1%), BGSA (2%), FS-pBGSK (1%), HGSA (1%), DGUFS (1%), and WOASAT (2%). For the Soyabean dataset, the proposed method BRCSA is competitive with all other existing algorithms in the literature regarding classification accuracy. For the Lung Cancer dataset, the suggested algorithm performs better than the current algorithms DFS (7%), MBOICO (1%), MBOLF (1%), FS-BGSK (5%), BGSA (4%), FS-pBGSK (2%), HGSA (5%), and WOASAT (3%). Additionally, it rivals DGUFS in terms of categorization precision. For the Soyabean dataset, the suggested approach outperformed DFS (4%), FS-BGSK (2%), BGSA (4%), FS-pBGSK (2%), HGSA (2%), DGUFS (1%), and WOASAT (5%). In terms of categorization accuracy, it is on par with MBOICO and MBOLF.

Discussion

This section provides information on the number of multimodal solutions in the binary external archive. Moreover, to validate the efficacy of the proposed model, we tried to find multiple solutions to a single problem without sacrificing the best system performance. For the proposed BRCSA, Table 5 and Fig. 3 shows the total number of multimodal solutions obtained on the binary novel archive and the other algorithms’ final population set. This table compares the total number of unique solutions produced across six datasets. The six datasets include Sonar, Glass, Wdbc, Protein, Segment, and Vehicle. Tables 6 and Fig. 4 provides the average fitness value (AFIT) of proposed method with other compared methods. Table 7 and Fig. 5 illustrates the average classification accuracy (ACA) obtained by proposed model and other existing models. With many distinctive traits on their end of the population, algorithms like WOASAT, DGUFS, and BSSA variations perform well compared to Table 6 results. On all other datasets besides the Segment dataset, MO techniques perform better than the current algorithms.

Table 5 Comparison results on the number of RCSA feature with existing algorithms.

NoFSub	Glass	WBDC	Sonar	Vehicle	Protein	Segment	
DFS	5	12	3	6	5	4	
MBOICO	3	S	9	8	6	4	
MBOLF	2	2	15	6	12	3	
FS-BGSK	7	7	4	8	8	11	
BGSA	6	10	8	6	8	11	
FS-pBGSK	6	7	8	8	5	11	
HGSA	3	6	4	5	6	8	
DGUFS	1	4	4	2	3	2	
WOASAT	3	1	2	3	2	1	
BRCSA	2	1	2	2	2	1	
BSSA	3	7	4	7	6	1	

Figure 3 Comparison of the number of features for various instances.

Table 6 Results on the average fitness of RCSA vs existing algorithms.

AvgFit	Glass	WBDC	Sonar	Vehicle	Protein	Segment	
DFS	0.6765	0.8456	0.8894	0.6978	0	1	
MBOICO	0.7071	0.9391	0.9182	0.7342	0.7859	0.9335	
MBOLF	0.7253	0.9493	0.9252	0.7317	0.7506	0.9314	
FS-BGSK	0.7448	Q9319	0.9038	0.7373	0.7793	0.2275	
BGSA	0.7135	0.9462	0.9111	0.7338	0.7753	0.9289	
FS-pBGSK	0.6909	0.9069	0.8617	0.7108	0.6885	0.908	
HGSA	0.7071	0.9426	0.9179	0.7428	0.7983	0.9313	
DGUFS	0.7217	0.9356	0.9243	0.7417	0.778	0.9348	
WOASAT	0.7198	0.9224	0.8836	0.7172	0.7732	0.939	
BRCSA	0.7321	0.9213	0.8873	0.7297	0.8063	1	
BSSA	0.7128	0.931	0.9045	0.7345	0.7781	0.939	

Figure 4 Comparison of average features for different instances.

Table 7 Results on average classification accuracy of RCSA vs exiting algorithms.

AvgCA	Glass	WBDC	Sonar	Vehicle	Protein	Segment	
DFS	0.5621	0.2472	0.9634	0.6063	0.7363	0.9092	
MBOICO	0.6078	0.9286	0.9752	0.6507	0.7405	0.9264	
MBOLF	0.6255	0.9454	0.9845	0.6487	0.7043	0.9234	
FS-BGSK	0.65813	0.9241	0.9697	0.6586	0.7481	0.9223	
BGSA	0.6120	0.9438	0.9658	0.6506	0.7433	0.9249	
FS-pBGSK	0.7093	0.9302	0.9196	0.6261	0.7279	0.963	
HGSA	0.6078	0.9369	0.9505	0.6682	0.7535	0.9285	
DGUFS	0.6167	0.9312	0.9668	0.6562	0.7209	0.9283	
WOASAT	0.6235	0.9248	0.948	0.6270	0.7224	0.9342	
BRCSA	0.764	0.9483	0.9871	0.729	0.784	0.928	
BSSA	0.6135	0.9259	0.9457	0.6503	0.7258	0.9342	

Figure 5 Comparison of the average accuracy for various situations.

For the Glass dataset, the suggested approach outperforms the currently used algorithms, including DFS at 60%, MBOICO at 33%, FS-BGSK at 71%, BGSA at 67%, FS-pBGSK at 67%, HGSA at 33%, WOASAT with 33%, and it is competitive with MBOLF. DGUFS, however, significantly outperforms the suggested method. For the Wbdc dataset, the proposed approach performs better than the current algorithms, including DFS (92%), MBOICO (88%), MBOLF (50%), FS-BGSK (86%), BGSA (90%), FS-pBGSK (86%), HGSA (83%), and DGUFS (75%), and is on par with WOASAT. For the sonar dataset, the suggested method outperformed DFS at 33%, MBOICO at 78%, MBOLF at 87%, FS-BGSK at 50%, BGSA at 75%, FS-pBGSK at 75%, HGSA with 50%, and DGUFS with 50%, and it is on par with WOASAT.

For the Vehicle dataset, the proposed algorithm outperformed DFS with a score of 67%, MBOICO with a score of 75%, MBOLF with a score of 67%, FS-BGSK with a score of 75%, BGSA with a score of 67%, FS-pBGSK with a score of 75%, HGSA with a score of 60%, WOASAT with a score of 33%, and DGUFS on a par. For the Protein dataset, the proposed method performs better than the existing algorithms DFS with 60%, MBOICO with 67%, MBOLF with 83%, FS-BGSK with 75%, BGSA with 75%, FS-pBGSK with 60%, HGSA with 67%, DGUFS with 33%, and is on par with WOASAT.

For the segment dataset, the proposed method outperformed DFS with a performance of 75%, MBOICO with a commission of 75%, MBOLF with a performance of 67%, FS-BGSK with a performance of 91%, BGSA with a performance of 88%, DGUFS with 50%, and WOASAT with a performance of 91%. For the Glass dataset, the proposed approach performs better than the current algorithms DFS (8%), MBOICO (3%), MBOLF (1%), BGSA (3%), FS-pBGSK (6%), HGSA (3%), DGUFS (1%), and WOASAT (2%). But the FS-BGSK algorithm performs better than the current algorithm by 2% on the average fitness value.

For the Wbdc dataset, the proposed approach performs better than other algorithms like DFS (8%), FS-pBGSK (2%), and WOASAT (competing equally well). However, regarding average fitness value, FS-BGSK, MBOICO, MBOLF, FS-BGSK, BGSA, HGSA, and DGUFS beat the current method. For the Protein dataset, the new approach outperforms the existing algorithms DFS, MBOICO, MBOLF, FS-BGSK, BGSA, FS-pBGSK, HGSA, DGUFS, and WOASAT by a factor of 4% each. For the Segment dataset, the proposed algorithm outperformed MBOICO, MBOLF, FS-BGSK, BGSA, FS-pBGSK, HGSA, DGUFS, WOASAT, and DFS by a combined margin of 7%, 7%, 4%, 4%, and 15%, respectively. The performance of the proposed BRCSA is not above average in terms of average fitness value for datasets like Vehicle and Sonar.

For the Glass dataset, the suggested method performs better than the already used algorithms DFS (26%), MBOICO (20%), MBOLF (18%), FS-BGSK (14%), BGSA (20%), FS-pBGSK (7%), HGSA (20%), DGUFS (19%), and WOASAT (18%). The proposed approach outperforms the existing algorithms for the Wbdc dataset, including DFS (74%), MBOICO (2%), FS-BGSK (3%), FS-pBGSK (2%), HGSA (1%), DGUFS (2%), WOASAT (2%), and competes on an equal footing with BGSA and MBOLF. On the sonar dataset, the proposed approach outperformed the following algorithms: DFS (2%), MBOICO (1%), MBOLF (0%), FS-BGSK (2%), BGSA (2%), FS-pBGSK (7%), HGSA (4%), DGUFS (2%), and WOASAT (4%).

For the Vehicle dataset, the proposed algorithm outperformed DFS (17%), MBOICO (11%), MBOLF (11%), FS-BGSK (9%), BGSA (11%), FS-pBGSK (14%), HGSA (8%), DGUFS (10%), and WOASAT (14%). For the Protein dataset, the proposed approach beats the existing algorithms DFS, MBOICO, MBOLF, FS-BGSK, BGSA, FS-pBGSK, HGSA, DGUFS, and WOASAT by 6%, 5%, 7%, 8%, and 10%, respectively. For the Segment dataset, the proposed algorithm outperformed DFS by 2%, FS-BGSK by 1%, and BGSA, MBOICO, MBOLF, HGSA, and DGUFS on par. However, with 4% and 1% improvements above BRCSA, respectively, FS-pBGSK and WOASAT outperform them.

Time complexity analysis of binary reinforced cuckoo search algorithm

The time complexity of the proposed BRCSA lies on three major parameters, and they are the size of the population (N),number of decision variables (d),number of iterations ( Imax). The time complexity of BRCSA has been evaluated phase-wise. In each phase, the time complexity will be represented in asymptotic notation. In the end, the step-wise time complexity values can be summed up, and the final deal will be described in the asymptotic form. i) Population initialization phase: The individuals in the population will be initialized, and the initialization will highly depend on the population size ( N) and decision variables ( d). The population initialization will be carried out by initializing every solution variable. Hence the time complexity can be computed as O(N×d).

ii) Generation of next iteration solution: This phase has been divided further into different sub-categories as follows: Random solution generation: two solution initializations will take the time complexity as (2×d) and it will be carried out for all N solutions; hence the time complexity is O(N×2.d)=O(N×d)

Replacement of best individual: In this phase, the fitness of every function will be compared with the respective newly generated solution, and the best one shall be replaced. This phase will take the time complexity as O(N).

Unit value generation: From lines 21 through 26, all take the time complexity of O (1) and let us consider it as constant c.

Patron-Prophet Phase: This number is highly dependent on the number of abandoned solutions and, considering the upper bound, assumes that half of the solutions are abandoned. In this case, the time complexity for the generation of new solutions will be O(N2×d).

On consolidating all the sub-categories of 2nd phase, it will be O(N×d)+O(N)+O(N2×d)+c=O(N×d).

iii) The 2nd phase will be iterated for a maximum of the total number of iterations (Imax). Hence the total time complexity will be O(Imax×N×d).

On the whole process, the total computation of Patron-Prophet can be summarized as T(BRCSA)=O(Imax×N×d)+O(N×d). Asymptotic notations considering the upper bound time complexity can be represented as T(BRCSA)=O(Imax×N×d).

Conclusion

The Binary Reinforced Cuckoo Search Algorithm (BRCSA) for Multimodal Feature Selection is a promising approach for selecting relevant features in high-dimensional data sets. The BRCSA algorithm is a modified version of the Cuckoo Search Algorithm (CSA) that incorporates binary reinforcement to enhance the search ability of the algorithm. By combining the BRCSA algorithm with a classification method, the proposed approach can effectively select the most relevant features for classification tasks. Eleven distinct datasets have been used to evaluate the suggested strategy for the feature selection problem. The experimental results show that the proposed approach outperforms other state-of-the-art feature selection methods regarding classification accuracy and computational efficiency. The BRCSA-based feature selection method also demonstrates better robustness against noise and outliers than other methods. However, the time complexity in BRCSA becomes a challenge when the dataset size increases. As the number of features increases, the complexity of the algorithms can increase exponentially, leading to scalability issues. And another limitation is there is no significant method to distinguish between highly correlated features. In conclusion, the BRCSA-based feature selection approach is a promising and effective method for selecting relevant features in high-dimensional data sets for classification tasks. It has potential applications in various domains, such as bioinformatics, image processing, and text mining. The future direction of the proposed method is to tune the proposed multimodal based feature selection towards solving multi-objective-based feature selection algorithm with the intuition of search space aware techniques.

Additional Information and Declarations

Competing Interests

Author Contributions

Data Availability

The authors declare that they have no competing interests.

Kalaipriyan Thirugnanasambandam conceived and designed the experiments, performed the experiments, performed the computation work, prepared figures and/or tables, authored or reviewed drafts of the article, and approved the final draft.

Jayalakshmi Murugan conceived and designed the experiments, performed the experiments, performed the computation work, authored or reviewed drafts of the article, and approved the final draft.

Rajakumar Ramalingam performed the experiments, performed the computation work, prepared figures and/or tables, authored or reviewed drafts of the article, and approved the final draft.

Mamoon Rashid analyzed the data, prepared figures and/or tables, authored or reviewed drafts of the article, and approved the final draft.

R. S. Raghav analyzed the data, performed the computation work, prepared figures and/or tables, and approved the final draft.

Tai-hoon Kim analyzed the data, authored or reviewed drafts of the article, and approved the final draft.

Gabriel Avelino Sampedro performed the experiments, authored or reviewed drafts of the article, and approved the final draft.

Mideth Abisado conceived and designed the experiments, authored or reviewed drafts of the article, and approved the final draft.

The following information was supplied regarding data availability:

The code is available at GitHub and Zenodo:

- https://github.com/ramukshare/Feature-Selection.

- ramukshare. (2023). ramukshare/Feature-Selection: FS (V1.0). Zenodo. https://doi.org/10.5281/zenodo.10425716.

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
