# Peer review of "Optimizing multimodal feature selection using binary reinforced cuckoo search algorithm for improved classification performance"

_PeerJ Computer Science, doi:10.7717/peerj-cs.1816_

## Round 0.1 · original submission · Major Revisions

Dear authors,

Thank you for submitting your article. Reviewers have now commented on your article and suggest major revisions. When submitting the revised version of your article, it will be better to address the following:

1- Please include future research directions.

2- Present strong discussions leading to the different aspects of metaheuristic approaches based feature selection.

3- The values for the parameters of the algorithms selected for comparison are not given.

4- The paper lacks the running environment, including software and hardware. The analysis and configurations of experiments should be presented in detail for reproducibility. It is convenient for other researchers to redo your experiments and this makes your work easy acceptance. A table with parameter settings for experimental results and analysis should be included in order to clearly describe them.

5- The authors should clarify the pros and cons of the methods. What are the limitation(s) methodology(ies) adopted in this work? Please indicate practical advantages, and discuss research limitations.

6- The research gaps and contributions should be clearly summarized in the introduction section. Please evaluate how your study is different from others in the related work section.

Reviewer 1 ·

Basic reporting

The authors proposed a novel optimization algorithm inspired by cuckoo birds' behaviour, the Binary Reinforced Cuckoo Search Algorithm (BRCSA). In addition, we applied the proposed BRCSA-based classification approach for multimodal feature selection. The proposed method aims to select the most relevant features from multiple modalities to improve the model's classification performance. The BRCSA algorithm is used to optimize the feature selection process, and a binary encoding scheme is employed to represent the selected features. However, they must attend to these comments to make the paper more quality for the reader and the reputable journal.
• The authors should improve on the abstract by providing quantitative results.
• The authors' in-text citations should be chronological. For instance, after citation [7], it should be [8], not [22] that I am seeing in line 101, page 3.
• Line 172, you don't cite with initial, so the initials J. and W. [25] should be removed.
• Line 175, this in-text citation H. et al. [26], should be corrected; it doesn't seem correct.
• In lines 189, 198, 203, 218, etc., the in-text citations should be corrected; that is, all initials in the citations should be removed.
• After the abstract, the authors should add keywords and give space before starting with the main article, "Introduction".
• In line 351, the author should include the data split in the datasets description. How much percentage of training, validation and testing set of the dataset is used to implement this study?
• How did the authors tune the optimal hyperparameter of all models? It should be described clearly.
• How did you solve the problem of an overfitting and small dataset
• The limitations of the study should be stated, and they should present future research work.
• Source codes should be provided for replicating the study
• The study should be compared with existing systems (state-of-the-art), and authors should state how it surpassed the existing one and why it performed less or less.
• There is a need for significant improvement in the general quality of the English language and the method in which it is presented. A substantial quantity of typographical errors and grammatical issues were observed. To ensure the linguistic accuracy and subject familiarity of your paper, it is advisable to engage the assistance of a colleague proficient in English and knowledgeable in the issue, or seek the services of a professional editing firm.
• We are in September 2023, and I could see just 1 2022 and no 2023 citations. Citing recent literature have several advantages for both authors and journal. It can help authors establish their credibility, demonstrate their research's relevance, and help avoid plagiarism. In the same way, it assists journals in increasing their visibility, improving their reputation, increasing their citation rates, and meeting reader expectations. Hence, for this reason, I have suggested some recent literature from 2023 and 2022 relating to the study that you are to cite and reference in your article.
a. Zheng, Y., Li, L., Qian, L., Cheng, B., Hou, W.,... Zhuang, Y. (2023). Sine-SSA-BP Ship Trajectory Prediction Based on Chaotic Mapping Improved Sparrow Search Algorithm. Sensors, 23(2), 704. doi: 10.3390/s23020704
b. Lu, C., Zheng, J., Yin, L., & Wang, R. (2023). An improved iterated greedy algorithm for the distributed hybrid flowshop scheduling problem. Engineering optimization. doi: 10.1080/0305215X.2023.2198768
c. Qian, L., Zheng, Y., Li, L., Ma, Y., Zhou, C.,... Zhang, D. (2022). A New Method of Inland Water Ship Trajectory Prediction Based on Long Short-Term Memory Network Optimized by Genetic Algorithm. Applied Sciences, 12(8), 4073. doi: 10.3390/app12084073
d. Zheng, Y., Lv, X., Qian, L., & Liu, X. (2022). An Optimal BP Neural Network Track Prediction Method Based on a GA–ACO Hybrid Algorithm. Journal of Marine Science and Engineering, 10(10), 1399. doi: 10.3390/jmse10101399
e. Wang, Y., Xu, N., Liu, A., Li, W., & Zhang, Y. (2022). High-Order Interaction Learning for Image Captioning. IEEE Transactions on Circuits and Systems for Video Technology, 32(7), 4417-4430. doi: 10.1109/TCSVT.2021.3121062
f. Li, W., Wang, Y., Su, Y., Li, X., Liu, A.,... Zhang, Y. (2023). Multi-Scale Fine-Grained Alignments for Image and Sentence Matching. IEEE Transactions on Multimedia, 25, 543-556. doi: 10.1109/TMM.2021.3128744
g. She, Q., Hu, R., Xu, J., Liu, M., Xu, K.,... Huang, H. (2022). Learning High-DOF Reaching-and-Grasping via Dynamic Representation of Gripper-Object Interaction. ACM Trans. Graph., 41(4). doi: 10.1145/3528223.3530091
h. Huang, W., Xia, J., Wang, X., Zhao, Q., Zhang, M.,... Zhang, X. (2023). Improvement of non-destructive detection of lamb freshness based on dual-parameter flexible temperature-impedance sensor. Food Control, 153, 109963. doi: https://doi.org/10.1016/j.foodcont.2023.109963
i. Lu, S., Ding, Y., Liu, M., Yin, Z., Yin, L.,... Zheng, W. (2023). Multiscale Feature Extraction and Fusion of Image and Text in VQA. International Journal of Computational Intelligence Systems, 16(1), 54. doi: 10.1007/s44196-023-00233-6
j. Qin X, Ban Y, Wu P, Yang B, Liu S, Yin L, Liu M, Zheng W. Improved Image Fusion Method Based on Sparse Decomposition. Electronics. 2022; 11(15):2321. https://doi.org/10.3390/electronics11152321

Experimental design

In line 351, the author should include the data split in the dataset's description. How much percentage or ration of training, validation and testing set of the dataset is used to implement this study?
The authors should state the configuration of the system that was used for experiment or implementation.
How did the authors tune the optimal hyperparameter of all models? It should be described clearly.
How did you solve the problem of an overfitting and small dataset

Validity of the findings

The author should state the validity of their study in a different section.

Reviewer 2 ·

Basic reporting

Abstract
1.The section clearly stated but the following should be clearly stated
2. The problem and objective of the research is missing. T
3. The method used should be properly describe to enable better understanding and improve the abstract
4. The The abstract should be concluded with fact from the experiment

Introduction
This section of the work is supposed to address the background to the work identifying the domain of the research work, establishing identified problem, objective of the work, arrangement of sections and conclude with the contribution. My observations are as follows:
1. The background did not thoroughly address the subject of discuss
2. The problem of discuss was not clearly stated
3. The objective of the work was not clearly stated
4. The section arrangement is missing
5. There is no flow of context and content of the introduction
Therefore, this section of the work should be reviewed to include the missing subjects to strengthen the introduction. Also each paragraph of the section must link to the next to ensure coherence and flow.

Structure
The structure was followed, the figures and tables was made available but he raw data was not shared.

Result

Experimental design

Research Question and objectives
Research problem and objectives was not clearly stated hence needs to be included into the work

The experiment shows high technical and ethical standard and the discussion of the result was clear enough

Methodology
Methodology section is missing. This section is very important to afford other researcher an opportunity to validate result by following the methodological approach used by the authors.
I suggest that this section be included in the work.

Validity of the findings

The results was valid as it shows improvement compared to existing work

Additional comments

The abstracts should be reviewed to include type of research, problem identified, objective of the research, method used, and conclude with the result based on facts.

The introduction section should be reviewed to include the missing subjects to strengthen the introduction. Also each paragraph of the section must link to the next to ensure coherence and flow

Methodology section is missing.. This section is very important to afford other researcher an opportunity to validate result by following the methodological approach used by the authors. I suggest that this section be included in the work.

The authors needs to include more relevant and recent work in the literature. This section must be thoroughly reviewed.

---

## Round 0.2 · Minor Revisions

Dear authors

We received the review reports for your revised manuscript. Thank you for addressing the issues raised by the reviewers. Some remaining concerns and issues that pertain to the in-text citations and a more comprehensive discussion of limitations. still seem to be addressed. We encourage you to address the concerns and criticisms of the reviewers and resubmit your article once you have updated it accordingly.

Best wishes,

**Language Note:** The review process has identified that the English language must be improved. PeerJ can provide language editing services - please contact us at copyediting@peerj.com for pricing (be sure to provide your manuscript number and title). Alternatively, you should make your own arrangements to improve the language quality and provide details in your response letter. – PeerJ Staff

Reviewer 1 ·

Basic reporting

Your efforts to address my feedback are greatly appreciated, and it is evident that you have made significant improvements to the manuscript. Upon reviewing the revised manuscript, it is clear that you have diligently worked on many of the points raised in our previous feedback. The paper now exhibits enhanced clarity and organization, significantly improving its quality.
However, a few remaining concerns still need to be addressed before I can proceed with accepting the manuscript. These remaining issues pertain to the in-text citations and a more comprehensive discussion of limitations.
I believe that addressing these remaining concerns will substantially strengthen your manuscript and ensure its alignment with the high standards of our journal. I kindly request your attention to these points in your next revision.
1. I want to appreciate the authors for including the recent and related articles suggested by me, but I noticed you have more than four citations in a sentence; this should be discouraged because of the following reasons:
• Putting many references in one line might be too much for certain readers. It may dilute the significance of the citations and make it more difficult for them to determine which sources are most relevant and essential.
• Diversion from the author's thoughts and analyses caused by excessive citations. The statement may read more like a collection of ideas from different scholars than a logical argument.
2. Hence, it is generally recommended to limit the number of citations in a single sentence and distribute them across multiple sentences or paragraphs as appropriate. This allows for better readability, focus, and clarity in your writing. Therefore, I suggest the author reduce each citation per sentence to four maximums. For instance, see pages 6-7, lines 232-233, lines 244 to 247, etc.
3. There are still a few grammatical errors and typos. I suggest you have a colleague proficient in English and familiar with the subject matter review your manuscript or contact a professional editing service.
4. It is essential to acknowledge the limitations of the study and engage in a discussion on potential avenues for future research.

Experimental design

No Comment

Validity of the findings

No Comment

Additional comments

No Comment

Reviewer 2 ·

Basic reporting

The problem and objective of the research have been included.
The abstract have been improved and concluded with facts from the experiment
The background have addressed the subject of discuss properly
The section arrangement included

Experimental design

The structure was followed, the figures and tables was made available but he raw data was not shared.
The method is described with details.

Validity of the findings

The results was valid as it shows improvement compared to existing work

---

## Round 0.3 · accepted · Accept

Dear authors,

Thank you for the revision. The paper seems to be improved in the opinion of the reviewers. The paper is now ready to be published.

Best wishes,

Reviewer 1 ·

Basic reporting

Your attention to detail and the precision with which you have implemented these changes are commendable.
I want to acknowledge your professionalism and dedication to refining your work, and I believe the revised manuscript will significantly benefit both the scientific community and future readers. Your openness to constructive criticism and your ability to translate it into tangible improvements make the peer review process valuable and rewarding.
Once again, thank you for your diligence and collaboration in ensuring the manuscript's quality.

Experimental design

No comment

Validity of the findings

No comment

Additional comments

No comment